# Sociocultural determinants of adoption of preventive practices for hantavirus: A knowledge, attitudes, and practices survey in Tonosí, Panama

Carlyn Harris[1], Blas Armién[1,2]*

**1** Department of Research in Emerging and Zoonotic Infectious Diseases, Gorgas Memorial Institute for Health Studies, Calle 35, Panamá, PANAMA, **2** Universidad Interamericana de Panamá, Panamá, PANAMA

\* barmien@gorgas.gob.pa

## Abstract

### Introduction

Hantaviruses are a group of single-stranded RNA viruses carried by small rodent reservoirs, transmitted to humans through inhalation of aerosolized particles of rodent feces, urine, or saliva. In Panama, the *Choclo orthohantavirus* has been associated with Hantavirus Pulmonary Syndrome (n = 54) and Hantavirus Fever (n = 53). In 2018, there were 107 cases of hantavirus diseases, the majority in the Tonosí district, and 4 deaths. As there is no vaccine or treatment for hantavirus, proper prevention measures by community members is key to stopping outbreaks.

### Methodology and principal findings

We investigated hantavirus knowledge, attitudes, and practices in one *corregimiento* of Tonosí, Panama to determine what factors influence uptake of prevention practices and high level of knowledge. We conducted a cross-sectional survey with 124 residents covering hantavirus knowledge, attitudes based in the Health Belief Model (perceived severity, perceived susceptibility, perceived obstacles, perceived benefits, and cues to action) and prevention practices. There was an overall high level of knowledge (median score: 4/6), though 20% did not know the route of transmission. The mean number of reported practices performed per person was 8.4 (range: 4–12). Most people had heard of hantavirus through other community members. In linear regression, lower perceived obstacles predicted higher preventive practice score. Reported obstacles to preventive practices included physical restrictions, such as age and health state. In ordinal logistic regression, higher education level and knowing more people who had previously been sick with hantavirus contributed to higher knowledge score.

### Conclusions

Future interventions should focus on removing barriers to performing preventive practices. As most people learned of hantavirus through community members, interventions should be

are available from the Gorgas Memorial Institute for Health Studies Institutional Ethics Committee (contact via combioetica@gorgas.gob.pa, 507.527.4823 or 507.527.4989) for researchers who meet the criteria for access to confidential data.

**Funding:** The work was financed by the Department of Research in Emerging and Zoonotic Diseases and by the project 111130150.501 (BA) from the Ministry of Economy and Finance of Panama. CH was supported by a Fulbright grant from the Fulbright U.S. Student Program, sponsored by the United States Department of State. BA (DI-UIP63380000) is supported by Research Direction, Universidad Interamericana de Panama. BA is a member of the SNI (Sistema Nacional de Investigación from SENACYT of Panamá). The funders had no role in study design, data collection and analysis, decision to publish, or preparation of the manuscript.

**Competing interests:** The authors have declared that no competing interests exist.

community-based and involve those who have experienced the disease. Any future education materials should address confusions about route of transmission and be targeted at those with a lower education level.

## Author summary

Hantavirus is a pathogen spread by small rodents in many regions of the world. In Panama, infection with hantavirus can lead to Hantavirus Fever or Hantavirus Pulmonary Syndrome. In 2018, there were 107 cases of hantavirus infection in Panama, the majority in the Tonosí District, and 4 deaths. Currently, there is no treatment, cure, or vaccine for hantaviruses. It is important that communities carry out the recommended prevention measures. In this study, we investigated what influences people to carry out the proper prevention measures and what influences people's knowledge of hantavirus in order to design interventions in highly affected communities. We found that barriers such as physical restrictions limit people's ability to perform the measures. Additionally, we found that those with higher education and those that knew more people who had been sick with hantavirus were more likely to have higher knowledge of the disease. We recommend that future interventions are community-based and focus on removing obstacles to performing the recommended prevention practices and involve those who have been affected by the disease to spread information.

## Introduction

Hantaviruses are a group of single-stranded RNA viruses belonging to the Hantaviridae family. Over 30 types of hantavirus have been found to cause disease in humans around the world. Hantaviruses are carried by small rodent reservoirs and transmitted when humans inhale aerosolized particles of infected rodent feces, urine, or saliva. Infection with hantavirus can lead to two primary disease states: Hantavirus Fever with Renal Syndrome (HFRS) and Hantavirus Pulmonary Syndrome (HPS), sometimes referred to as Hantavirus cardiopulmonary syndrome (HCPS) [1]. The first outbreak of HFRS was described during the Korean conflict (1950–53), but the causative agent was not discovered until the late 1970s [2]. Strains of hantavirus causing HFRS have been found primarily in Asia and Europe. HPS was first described in 1993 in the Four Corners region of the United States, caused by the *Sin Nombre orthohantavirus* [3]. Since then, cases of HPS caused by various strains of hantavirus have been reported throughout North and Latin America.

HPS, the primary disease state in Latin America, presents with flu-like symptoms such as fever, chills, myalgia, and vomiting and can rapidly progress to respiratory failure and death in severe cases. To date, no specific treatments or vaccines have been developed for HPS and clinical management is largely supportive [1]. Because of this, prevention of hantavirus by minimizing human contact with rodents is key.

In Panama, *Choclo orthohantavirus* has been associated with HPS with a case-fatality rate of 16.5% [4]. The reservoir of *Choclo orthohantavirus* is *Oligoryzomys fulvescens*, the pygmy rice mouse. The first cases of hantavirus in Panama were reported in 1999–2000 in the Azuero region [5]. In 2018, there were 107 confirmed cases of hantavirus diseases in Panama and four deaths (case-fatality rate of HPS during 2018 was 7.5% [4/53]). 102 of these cases occurred in the Los Santos Province of the Azuero, the majority in the Tonosí District [6]. Preventive

measures to avoid human contact with rodents and their excrements are the most effective way to prevent infection with hantavirus. It is increasingly important that residents of endemic areas such as Tonosí are informed of the control and prevention measures. Since the discovery of hantavirus in the Azuero region in 2000, many educational campaigns have encouraged residents to carry out the prevention measures such as keeping their homes and surrounding areas free of trash, avoid accumulation of wood and other materials, and maintain food and grains in sealed containers [7]. However, cases of hantavirus continue to rise in the Tonosí district.

Conducting quantitative sociocultural studies such as those that assess knowledge, attitudes, and practices (KAP) of communities regarding hantavirus and hantavirus prevention strategies are an important step to designing education campaigns and interventions. KAP surveys help identify knowledge gaps, cultural beliefs, and perceptions that influence health behaviors for both infectious and chronic disease as well as barriers the community faces. The results of KAP surveys help create culturally-competent interventions that are specific to the affected community [8]. Recently, in Latin America, KAP surveys have been used to assess knowledge, attitudes, and practices and inform interventions for Dengue, Zika, Chikungunya, and Malaria [9–13].

Despite the growing number of hantaviruses cases worldwide, only a handful of published studies have explored the sociocultural aspects of the disease. In 2001, Chamorro et al. studied the effects of hantavirus education and prevention campaigns in Guarare, Las Tablas, and Santiago, Panama[7]. In 2004, McConnell conducted a multi-country study (involving a site in Tonosí, Panama) to determine the effects of hantavirus education campaigns on knowledge and practices of communities[14]. Additionally, two hantavirus KAP survey studies were performed in a rural community in Chile [15] and a cluster of Japanese communities in Argentina [16].

In order to design effective education campaigns and interventions, it is important to determine which sociocultural factors such as knowledge, perceptions, and attitudes influence preventive behavior and what barriers exist for the community. Such a study has not yet been conducted for hantavirus in Panama. The objective of this study is to use a knowledge, attitudes, and practices survey informed by the Health Belief Model to describe the current state of KAP among a highly effected community and determine which factors facilitate or impede the adoption of preventive measures against hantavirus and which influence knowledge. We hypothesized that at least one of the Health Belief Model elements (see below) and level of knowledge of hantavirus would be predictors of preventive practice. Additionally, we hypothesized that higher education level and higher income would predict knowledge score.

## Theoretical framework

The "attitudes" element of the KAP survey was informed by the Health Belief Model (HBM) [17]. This model states that a person will adopt (or not) preventive health practices based on:

- Perceived susceptibility (how susceptible a person feels they are to a disease)

- Perceived severity (how severe a person believes the disease to be)

- Perceived benefits (how well they feel preventive measures are at avoiding the disease)

- Perceived obstacles (what perceived barriers stop people from carrying out the preventive practices)

- Cues to action (factors that provoke preventive health behaviors/serve as reminders to carry out practices)

The Health Belief model states that if a person has high perceived susceptibility, high perceived severity, feel the recommended prevention actions are beneficial, feel little obstacles exist to do them, and are exposed to factors that provoke the prevention behavior, they will be more likely to practice the appropriate behavior. It should be noted that "Cues to action" is not considered an "attitude" but was listed above only to clarify how the survey was organized. Additionally, we informed our survey based on the original HBM by Rosenstock developed in 1942. A sixth element, "self-efficacy" was added later in the 1980s and is not often included in HBM studies [18]. It was omitted here for survey brevity. The Health Belief Model has been applied in both qualitative and quantitative methods and has been used to predict preventive behaviors and design health interventions for both chronic and infectious disease [19–24]. Another common model for predicting health behavior is the Theory of Planned Behavior (TPB) [25]. We primarily chose the HBM over the TPB because the TPB assumes that individuals have acquired resources and opportunities to perform the behavior when they have the intention to. The HBM takes into account obstacles to performing behaviors and is more appropriate in a resource-limited context such as our study area. The HBM has not yet been used to evaluate predictors of preventive practices for Hantavirus.

## Methods

### Ethics statement

This study was approved by the Institutional Bioethics Committee at the Gorgas Memorial Institute for Health Research (Instituto Conmemorativo Gorgas de Estudios de Salud).

### Study setting, sample, and recruitment

This cross-sectional community-based survey study was carried out in March 2019 in the *corregimiento* El Bebedero. This region of the Tonosí district was selected due to its high incidence of Hantaviruses cases. According to the 2010 census, El Bebedero had about 440 households (1330 residents) over 45 square miles. Due to time and resource constraints and to maximize sampling, six neighborhoods with the highest density of households in El Bebedero were chosen. These neighborhoods comprise 348 of the 440 households in El Bebedero. We calculated an initial sample size of 95 households based on a 95% confidence level, a ±10% margin for error and a response distribution of 50% (50% distribution requires largest sample) as the distribution of possible responses was unknown. To account for potential non-response and missing data, the sample goal was 120 completed surveys. One week before the survey, the community representative was contacted for approval and flyers were distributed to inform residents of our presence. Over a 3-day period, participants were recruited door-to-door and were selected spontaneously (i.e. the participants were not expecting the visit) through convenience sampling (i.e. we recruited those at home at the time of the survey). Every household in each neighborhood was approached. Because of time limitations, we were not able to return to homes that did not answer the door on our first attempt. We attempted to survey throughout the working day as well as the evening to maximize our response rate and gain representation from working and non-working household members. We sought to conduct a proportional amount of surveys per neighborhood population. For example, the largest neighborhood has 28% of the households in El Bebedero. We aimed to conduct about 35 of the surveys here. When this quota was met, we moved on to the next neighborhood to ensure representation of all areas of the *corregimiento*.

The inclusion criteria for participation was: 1) age 18 years or above and 2) at least one year living in the Tonosí District. The data collection team comprised five interviewers trained in the survey instrument.

If a resident expressed interest in participating, they read the informed consent and were allowed to ask questions. If the participant was not able to read, a witness read the informed consent to the participant. The participant, witness (if applicable), and interviewer all signed the consent form. No incentives were provided, and participation was completely voluntary. The survey was conducted in Spanish.

## Survey instrument

The 41-item questionnaire explored sociodemographic information, knowledge about hantavirus, hantavirus preventive practices, and the five elements of the Health Belief Model (perceived susceptibility, perceived severity, perceived benefits, perceived obstacles, and cues to action) which comprised the "attitudes" section. Additional questions explored information sources for hantavirus and how many people the participant knew who had contracted in the infection.

Knowledge of hantavirus was evaluated with questions on route of transmission, symptoms, and animal reservoir of the virus. For route of transmission, the following scores were assigned: correct response = 2 points, incorrect response = 0 points, partially correct response (mention of mouse reservoir but not full transmission route) = 1 point. For symptomology, scores were assigned based on the number of symptoms the participant could correctly name (5+ symptoms = 3 points, 3–4 symptoms = 2 points, 1–2 symptoms = 1 point, no symptoms = 0 points). Additionally, they were given 1 point if they correctly named the animal reservoir and 0 if they could not. The total score for this section was 6.

Perceived susceptibility and perceived severity were both evaluated with 6 Likert-type questions each. Each question was scored from 1 to 5, with 1 indicating the lowest perceived susceptibility or severity and 5 indicating the highest. There was also an "I don't know option" available so the participant did not feel forced to answer. For example, the question "How worried are you about getting ill with Hantavirus" had six possible responses: Not worried (1 point), A little bit worried (2 points), Neutral/Indifferent (3 points), Worried (4 points) and Very worried (5 points). The possible scores for perceived severity and perceived susceptibility were from 6 to 30. The sixth "I don't know" option was treated as missing data for the purposes of the analyses. For both perceived susceptibility and perceived severity, if a participant answered "I don't know" for 3 or more of the 6 questions in the scale, they were excluded from the analysis and were not given a final score for the section. If a participant answered "I don't know" for only 1 or 2 of the 6 questions in the scale, the missing values were replaced with the mean of the other values in the same scale.

The perceived benefits section consisted of 13 Likert items that evaluated how effective the participant believed a list of preventive practices to be (1–5 scale). Higher scores indicated higher perceived benefits (possible scores of 13–65). Perceived obstacles consisted of 12 Likert items (1–5 scale) to determine how difficult participants perceive each preventive practice to carry out. Higher scores indicated higher perceived obstacles (possible scores of 12–60). The cues to action section was evaluated using 4 "Yes" or "No" questions to determine how much the participant was exposed to cues that might trigger carrying out preventive measures. For example, "Other residents in my community remind me to carry out the preventive measures". "Yes" responses were scored as 1 and "No" with 0 (possible scores of 0–4).

Preventive practices were evaluated with the question: "What have you done to improve your home and surroundings to prevent hantavirus?". For each correct practice participants listed spontaneously, 2 points were given. Then, participants were prompted on a list of practices they had not spontaneously mentioned. If they responded "Yes" to performing this

practice, they were given 1 point. 1 point was given instead of 2 to account for possible response bias. A response of "No" was assigned 0 points. The maximum score for practices was 24.

### Data capture and analysis

Data was captured on tablets and uploaded in Epi Info 7 (Centers for Disease Control and Prevention). Data analysis was done in IBM Statistical Package for Social Science, version 24. Descriptive statistics were calculated for all sociodemographic characteristics, knowledge, and for each element of the Health Belief Model. Due to many variables not following a normal distribution, a Spearman correlation was used to explore correlations between the continuous variables such as practice score, knowledge score, scores for elements of the Health Belief model and selected sociodemographic variables.

To identify predictors of practice score among the population, a linear regression was used. All HBM elements scores and knowledge score were entered in the model with selected sociodemographic characteristics that were found to be correlated with practice score in the Spearman correlation matrix. The model was iterated to maximize the adjusted R Square value. To identify sociodemographic predictors of knowledge score among the participants, an ordinal logistic regression was used. This model type was used because the knowledge score had a limited number of possibilities (integers 0–6) thus did not satisfy the assumptions for a linear regression model.

## Results

### Sample characteristics

Table 1 presents the sociodemographic information of the participants included in the study. Of the 135 households targeted for the study, 124 households participated (response rate: 92%). Of the 124 participants, 79 (63.7%) were female. The median age was 55 (IQR:36,70), with males being slightly older than females. The median year in school completed was year 6 (IQR: 5,9), or the end of primary school. 18 (14.5%) had been sick previously with Hantavirus. However, respondents reported knowing a median of 5 people (IQR: 2,8) who had suffered the illness. Most participants were housewives (n = 65, 52.4%) and 27 (21.8%) were farmers. Reported median family monthly income was $200 (IQR: 120, 350).

### Hantavirus knowledge

All participants (n = 124) had heard of hantavirus. Most participants (n = 91/124, 73.4%) had heard of hantavirus by word-of-mouth (neighbors, friends, family) during times that cases were reported in the community, compared to 18/124 (14.5%) who heard through the television and 18/124 (14.5%) through health care professionals. 60.5% of respondents correctly identified the route of transmission in entirety, with 10.5% recognizing the mouse reservoir but not naming the full transmission route. 20.2% did not know the transmission route and 6.5% believed hantavirus was caused by local crop fumigations near homes. 85.4% of participants could name at least one of the symptoms correctly. The most commonly cited symptoms were fever and chills (78%), headache (58.9%) and general pains (39.5%). When asked what animal transmits the virus, 86% answered correctly. It should be noted that the 86% correct for naming the animal differs from the 71% (60.5% + 10.5%) above for naming the mouse as a part of the route of transmission. Further details on responses for all survey questions can be found in the supporting information (S1 SurveyData).

**Table 1. Sociodemographic characteristics of the sample (n = 124).**

| | N (124) (%) | Median (Interquartile range) | | |
| --- | --- | --- | --- | --- |
| | | Total | F | M |
| **Gender** | | | | |
| Female | 79 (63.7) | | | |
| Male | 45 (36.3) | | | |
| Age | | 55 (36,70) | 48 (32,69) | 60 (44.5,70.5) |
| 18–34 | 30 (24.2) | | | |
| 35–50 | 28 (22.6) | | | |
| 51–67 | 26 (21.0) | | | |
| 68+ | 40 (32.3) | | | |
| Education (years) | | 6 (5,9) | 6 (3.5, 8) | 6 (6,11) |
| Primary (1–6) | 64 (51.6) | | | |
| Secondary (7–12) | 35 (28.2) | | | |
| University+ (13+ years) | 13 (10.5) | | | |
| Family income/month ($) | | 200 (120,350) | | |
| Time living in the area (years) | | 26.5 (12.25, 48) | | |
| Number of people known who have had hantavirus | | 5 (2,8) | | |
| Had hantavirus? | | | | |
| Yes | 18 (14.5) | | | |
| No | 105 (84.7) | | | |
| Profession | | | | |
| Ama de casa/housewife | 65 (52.4) | | | |
| Farmer | 27 (21.8) | | | |
| Sales/Independent | 7 (5.6) | | | |
| Professional | 4 (3.2) | | | |
| Technical worker | 2 (1.6) | | | |
| Construction worker | 3 (2.4) | | | |
| Student | 2 (1.6) | | | |
| Unemployed | 4 (3.2) | | | |
| Other | 9 (7.3) | | | |
| Community | | | | |
| Buenos Aires | 18 (14.5) | | | |
| El Bebedero | 41 (33.1) | | | |
| La Bonita | 16 (12.9) | | | |
| La Cacica | 25 (20.2) | | | |
| La Corocita | 10 (8.1) | | | |
| Perina | 14 (11.3) | | | |

## Preventive practices

Accounting for both spontaneous and prompted responses, the preventive practices which the highest number of people perform were cleaning and taking out trash (n = 124, 100%), picking up accumulates of waste materials, wood, etc. around the home (n = 113/124, 91%), mopping the home (n = 113/124, 91%), keeping food and water in contained with lids and away from rodents (n = 113/124, 91%), and keeping grains in sealed containers (n = 105/124, 84.7%). However, only 20% said they used gloves while cleaning, about half said they used a mask when cleaning, and 56% said they kept their trash in rodent-resistant receptacles. Fig 1 also describes percentage of participants performing each preventive practice. Table 2 shows below the preventive practices, the number of people who said they did them spontaneously, and the

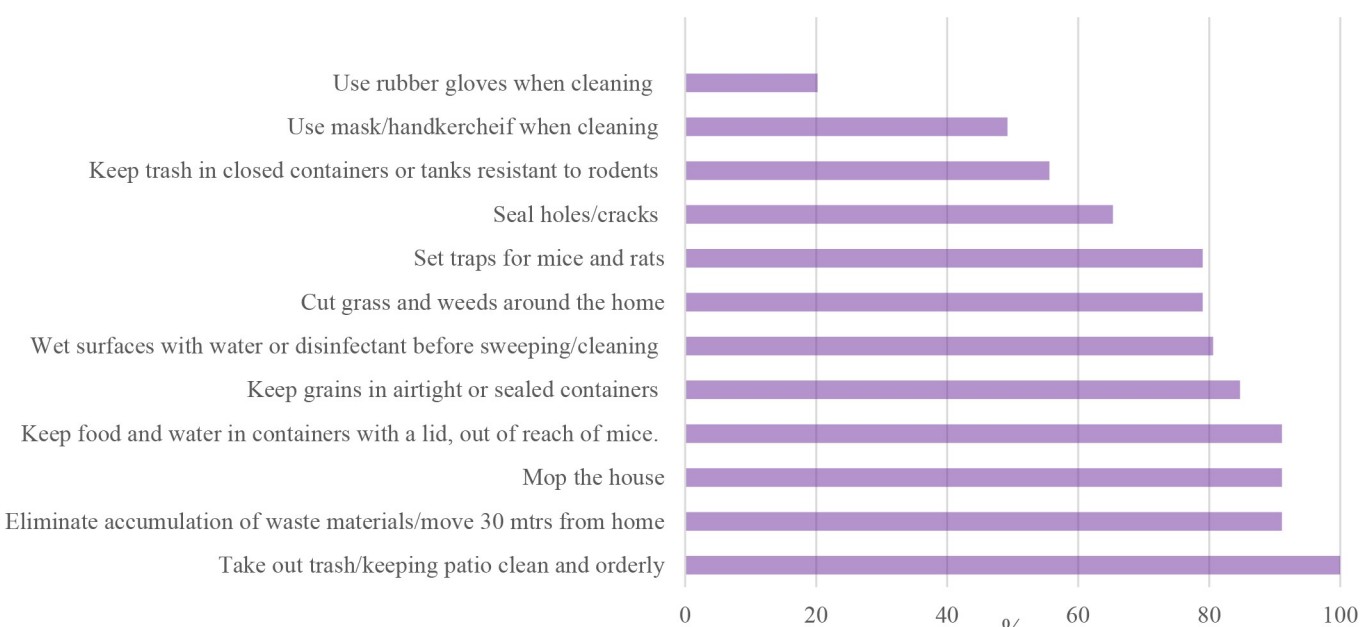

**Fig 1. Graphical representation of Table 2.** Percentage of participants performing each preventive measure (sum of spontaneous and after prompting responses).

total who said they performed the practice, accounting for spontaneous and prompted answers. "Cats" was not a part of the original list, so did not have a prompted option. Table 3 shows the frequency of participants performing different numbers of preventive actions. Accounting for both spontaneous and prompted responses, the mean number of practices carried out per person was 8.4 (SD: 1.70, range 4–12). All participants reported that they performed at least 4 of the 12 preventive practices. About 87% (n = 108) of participants reported performing 7 or more of the 12 preventive practices.

**Table 2. Percent of respondents carrying out preventive practices.**

| Preventive practice | Spontaneous answers | | Spontaneous + after prompting | |
|---|---|---|---|---|
| | N | % (n = 124) | N | % (n = 124) |
| Take out trash/keeping patio clean and orderly | 114 | 91.9 | 124 | 100.0 |
| Eliminate accumulation of waste materials and move accumulations of wood/forages/bales a minimum of 30 meters away from the home | 42 | 33.9 | 113 | 91.1 |
| Cut grass and weeds around the home | 7 | 5.6 | 98 | 79.0 |
| Set traps for mice and rats | 7 | 5.6 | 98 | 79.0 |
| Use mask, handkerchief, other to cover your mouth and nose when cleaning | 3 | 2.4 | 61 | 49.2 |
| Use rubber gloves when cleaning | 1 | 0.8 | 25 | 20.2 |
| Wet surfaces with water or disinfectant before sweeping/cleaning | 23 | 18.5 | 100 | 80.6 |
| Mop the house | 9 | 7.3 | 113 | 91.1 |
| Keep trash in closed containers or tanks resistant to rodents | 8 | 6.5 | 69 | 55.6 |
| Keep grains in airtight or sealed containers | 37 | 29.8 | 105 | 84.7 |
| Seal holes/cracks | 1 | 0.8 | 81 | 65.3 |
| Keep food and water in containers with a lid, out of reach of mice. | 43 | 34.7 | 113 | 91.1 |
| Cats (as predators) | 18 | 14.5 | | |

**Table 3. Frequency of participants performing recommended preventive practices.**

| Number of practices performed | Frequency (%) (n = 124) |
|---|---|
| 4 | 3 (2.4) |
| 5 | 4 (3.2) |
| 6 | 9 (7.3) |
| 7 | 21 (16.9) |
| 8 | 27 (21.8) |
| 9 | 26 (21) |
| 10 | 22 (17.7) |
| 11 | 11 (8.9) |
| 12 | 1 (0.8) |

## Health belief model

**Perceived susceptibility.** 63% (78/124) of respondents said they were worried or very worried about getting sick with hantavirus, and 80% (100/124) were worried or very worried that their family would get sick with hantavirus. About 15% (19/124) thought they were less susceptible than other people of their same age group to get ill, and 31% (39/124) said other groups in their community had a higher risk of getting hantavirus than them.

**Perceived severity.** 94% (117/124) of respondents said that hantavirus was dangerous or very dangerous and 78.2% (97/124) said it is very difficult to recover from the disease. When asked how severe the disease would be if they themselves got sick with hantavirus, 55% (68/124) said severe or very severe and 30% (38/124) weren't sure. About 47% of respondents said it was "probable" or "very probable" that they would die if they became ill with hantavirus, while 35.5% did not know what their fate would be.

**Perceived benefits.** For almost all 12 preventive practices we asked respondents about, over 80% said that the practices were either effective or very effective. About 16% (20/124) of respondents said that setting traps for rodents was either not effective or only slightly effective. Additionally, about 60.5% (75/124) agreed or strongly agreed that if they adopted all the prevention measures, they would not get sick. However, still 20% did not think that adopting the prevention measures would help them avoid the disease.

**Perceived obstacles.** Most reported all the recommended preventive practices to be easy or very easy to carry out. There were only 3 practices for which more than 10% (12+ of 124) of respondents reported difficulty performing: taking out the trash and keeping the home/patio clean and orderly (n = 30/124, 24.2%), removing accumulations of waste materials, wood etc. away from the home (n = 19/124, 15.3%), and using gloves when cleaning (n = 14/124, 11.3%). When asked why, almost half of respondents said it was difficult to keep up with general cleaning and trash maintenance (14/30) due to physical restrictions such as age or health reasons. Of those who reported using gloves difficult, 4/14 said it was due to cost and 4/14 said using them was uncomfortable. When asked why they thought other people in their community did not perform the recommended preventive measures, 38/124 (30.6%) said because they just did not feel like doing them. 35/124 others (28.2%) said others did not perform the measures because they do not believe in hantavirus or do not believe mice transmit the virus, but instead blame crop fumigation as the cause of the disease.

**Cues to action.** 65% of respondents said they have heard others in their community talk about hantavirus, and about half said they other residents in their community remind them to carry out the prevention measures. 67% of respondents said that community leaders, doctors, and government authorities remind them to carry out the prevention measures. Almost all

participants (91%) said they had seen or heard messages about hantavirus and the prevention measures on television, radio, or through flyers.

### Information sources

69% (85/124) of participants reported they knew where to receive information about hantavirus if they wanted it—most of them said from the hospital or a health center (67/124,78.8%). 86% (107/124) said they trust health care professionals to receive health information. The top three mediums from which respondents would prefer to receive information about hantavirus were television (n = 29/124, 23.4%), household visits (n = 29/124, 23.4%), and from health centers (n = 22/124, 17.7%). Additionally, 93.5% (116/124) of respondents knew at least one person who had been infected with hantavirus, with a median of 5 people (IQR: 2,8).

### Knowledge, practices, and the health belief model scores and correlations

Table 4 illustrates the median/mean point values for knowledge, the HBM elements, and preventive practices. The median total knowledge score (out of 6) was 4 (IQR:2.25, 5) and mean score for preventive practices (out of max 24) was 10.7 with 95% CI (10.3, 11.1). The Mann-Whitney U test revealed significant (p = 0.03) differences in knowledge scores between males [median (IQR): 4 (2,4)] and females [median (IQR): 4 (3,5)]. Though the median scores of these groups are the same, the interquartile range for females is slightly higher. A t-test did not reveal significant differences between males and females for practice scores. Additionally, no significant differences were found between those who had previously suffered hantavirus vs. those who had not for knowledge, HBM elements, or practice score. Table 5 shows Spearman correlations between practice scores, knowledge scores, the elements of the health belief model, and sociodemographic variables of interest. Practice score was weakly positively correlated with education (0.31, p<0.01) and income (0.31, p<0.01) and weakly negatively correlated with perceived obstacles score (-0.25, p<0.05) and age (-0.32, p<0.01). Knowledge score was weakly correlated with susceptibility score (0.28, p<0.01), number of people known who had hantavirus (0.35, p<0.01) and moderately positively correlated with education (0.5, p<0.01) and monthly income (0.45, p<0.01).

**Predictors of preventive practices.** The final multiple linear regression (Table 6) model explained 21.9% of the variation in preventive practice score (p = 0.004). Controlling for other variables such as age, years of education, and income, perceived obstacles score was the only variable found to make an independent statistically significant contribution (Beta = -0.21, p = 0.03). The model demonstrates that practice score decreases by about 0.2 units for every unit increase in perceived obstacles score (95% CI: -0.37, -0.02). 98 participants were included

**Table 4. Knowledge, practice, and HBM elements score.**

|  | Range possible | Points (median) | Interquartile range |
|---|---|---|---|
| **Knowledge** | 0–6 | 4 | 2.25, 5 |
| **Perceived susceptibility** | 6–30 | 21 | 18,24 |
| **Perceived severity** | 6–30 | 24 | 21,25 |
| **Perceived obstacles** | 12–60 | 25 | 24,28 |
| **Perceived benefits** | 13–65 | 52 | 49,52 |
| **Cues to action** | 0–4 | 3 | 2,4 |
|  |  | **Points (mean)** | **95% Confidence interval** |
| **Practices** | 0–24 | 10.7 | (10.3, 11.1) |

**Table 5. Spearman correlation of practice score, knowledge score, HBM elements, and select sociodemographic variables (\*p<0.05, \*\*p<0.01).**

|  | 1 | 2 | 3 | 4 | 5 | 6 | 7 | 8 | 9 | 10 | 11 | 12 |
|---|---|---|---|---|---|---|---|---|---|---|---|---|
| Practice score (1) | 1 | | | | | | | | | | | |
| Knowledge Score (2) | 0.156 | 1 | | | | | | | | | | |
| Susceptibility Score (3) | 1.27 | 0.281\*\* | 1 | | | | | | | | | |
| Severity Score (4) | -.101 | 0.025 | 0.17 | 1 | | | | | | | | |
| Obstacles score (5) | -0.248\* | -0.127 | -0.056 | -0.002 | 1 | | | | | | | |
| Perceived benefits score (6) | 0.051 | 0.063 | -0.124 | 0.249\* | -0.272\*\* | 1 | | | | | | |
| Cues to action (7) | 0.192 | 0.067 | 0.181 | -0.151 | -0.193 | -0.016 | 1 | | | | | |
| Age (8) | -0.320\*\* | -0.350\*\* | -0.291\*\* | -0.044 | 0.146 | 0.010 | -0.158 | 1 | | | | |
| Education (years) (9) | 0.308\*\* | 0.499\*\* | 0.189 | -0.075 | -0.026 | -0.035 | 0.089 | -0.552\*\* | 1 | | | |
| Income (10) | 0.313\*\* | 0.449\*\* | 0.216\* | -0.139 | -0.034 | -0.135 | 0.180 | -0.415\*\* | 0.432\*\* | 1 | | |
| Years living in area (11) | -0.197 | -0.192 | -0.020 | -0.098 | 0.099 | -0.120 | 0.049 | 0.550\*\* | -0.318\*\* | -0.265\*\* | 1 | |
| # of people know who have had Hanta (12) | 0.134 | 0.348\*\* | 0.195 | -0.016 | -0.172 | -0.150 | 0.173 | -0.085 | 0.176 | 0.145 | 0.040 | 1 |

in the multiple linear regression. 26 were not included as they had either a missing perceived susceptibility score or perceived obstacles score.

**Predictors of knowledge score.** The final ordinal logistic regression (Table 7) model to predict knowledge score had four independent variables (number of people known who have been sick with hantavirus, age, years of education, and gender). Income was originally included in the model but was removed as the parameter estimate was extremely close to 0 and not a significant contributor to the model. The full model was statistically significant (p<0.001) and explained between 37.8% (Cox and Snell R-Square) and 39.3% (Nagelkerke R-Square) of the variation in knowledge scores. Only two of the variables made a unique statistically significant contribution to the model: number of years of education and number of people who the respondent knew who had been sick with hantavirus. Years of education was the strongest predictor of knowledge score. With an odds ratio of 1.27 (95% CI: 1.14, 1.42), the model indicated that a one-year increase of education would increase the odds of having a higher hantavirus knowledge score by 1.27 times. Additionally, each increase in person known who was previously ill with hantavirus disease would increase odds of having a higher level of hantavirus knowledge by 1.1 times (1.02, 1.18).

## Discussion

The objective of this study was to evaluate hantavirus knowledge, attitudes (using the Health Belief Model) and preventive practices of a community in Tonosí, Panama and determine

**Table 6. Multiple linear regression, n = 98 (dependent variable: practice score).**

|  | Unstandardized Coefficients | | | Standardized Coefficients | | |
|---|---|---|---|---|---|---|
|  | **B** | **95% CI (B)** | **Std. error** | **Beta** | **t** | **Sig.** |
| **Constant** | 16.66 | 8.76, 24.56 | 3.97 | | 4.13 | 0.000 |
| **Perceived obstacles** | -0.19 | -0.37, -0.02 | 0.09 | -0.210 | -2.19 | 0.031 |
| **Cues to action** | 0.20 | -0.19, 0.59 | 0.19 | 0.10 | 1.02 | 0.312 |
| **Perceived benefits** | 0.04 | -0.04, 0.12 | 0.40 | 0.10 | 1.02 | 0.311 |
| **Perceived severity** | -0.14 | -0.30, 0.02 | 0.04 | -0.17 | -1.77 | 0.081 |
| **Perceived susceptibility** | 0.009 | -0.11,0.13 | 0.06 | 0.06 | 0.16 | 0.131 |
| **Age** | -0.02 | -0.05, 0.01 | 0.01 | -0.18 | -1.57 | 0.119 |
| **Years of education** | 0.11 | -0.04, 0.26 | 0.08 | 0.19 | 1.45 | 0.152 |
| **Monthly family income** | 0.000 | -.002,0.001 | 0.001 | 0.02 | 0.16 | 0.877 |

**Table 7. Ordinal logistic regression (dependent variable: knowledge score).** Odds ratios refer to the odds of obtaining a higher knowledge score, compared to a lower score, so OR >1 implies increased odds (n = 124).

|  | Odds ratio | 95% Confidence interval | Sig. |
|---|---|---|---|
| **Number of people known who have been sick with hantavirus** | 1.10 | 1.02, 1.18 | 0.006 |
| **Level of education** | 1.27 | 1.14, 1.42 | 0.000 |
| **Age** | 0.99 | 0.97, 1.01 | 0.169 |
| **Gender** | 0.71 | 0.36, 1.42 | 0.333 |

predictors of knowledge and preventive practices. All 124 participants had heard of hantavirus, mostly through word-of-mouth in the community and most knew at least one person who had been sick with hantavirus, with a median of 5 people. While the median knowledge score was 4 (IQR: 2.25, 5) of a possible 6, 20.2% of participants still reported not knowing the route of transmission of hantavirus. Additionally, this is the first published study to document the community belief that crop fumigations near homes are linked to hantavirus. A qualitative evaluation of this community concern would be useful. Future interventions should clarify the route of transmission and specifically address concerns about crop fumigations. Most participants could name the animal reservoir and at least one symptom. While the literature is scarce on this topic, McConnell conducted a multi-site (Chile, Panama, New Mexico) evaluation of education campaigns for hantavirus [14]. Part of their sample in Panama came from Tonosí and Pocri, two sites with documented hantavirus cases. In their sample, only 40.6% of participants had known someone with hantavirus. However, their data was collected in hospital and clinic waiting rooms, while the present study was conducted in a neighborhood with a high incidence of hantavirus, which accounts for this disparity. 93.2% of their respondents had heard of hantavirus and 91% knew the animal reservoir, similar to the 86% in our sample. Over 80% of participants in the McConnell study could name the correct transmission route [14], compared with 60% in our study. In the McConnell study and the present study, there seems to be a high overall knowledge, or at least familiarity, with the topic in Panama. Affected communities being exposed to educational messages about hantavirus since the first outbreak in Panama in 2000 likely explains the high familiarity. Almost all the respondents in our survey stated seeing or hearing hantavirus prevention messages on the radio, on television, or in flyers. Chomorro et al. also conducted a hantavirus KAP survey in Panama in 2001 after the first outbreak. Though it did not include respondents from Tonosí, other districts in the same province (Las Tablas and Guarare) were included. They found that 86% had heard of the disease and 60% knew the transmission route, similar to our study. However, 33% could not name any symptoms [7]. At least in terms of symptomology, knowledge has drastically increased since 2001. Another KAP study in rural Chile found that only 46% could name the route of transmission and only 45% of the sample knew someone with hantavirus [15], a much lower rate than what was found in this study. However, comparing these studies is difficult as the geographic spread of sampling frames and recruitment methods were vastly different.

The mean score for preventive practices was 10.7 of 24 (95% CI: 10.3,11.1). We note that to achieve a total score of 24, a respondent would have had to spontaneously list 12 cleaning practices as spontaneous answers were assigned a higher point value than prompted answers. Overall, the community reports a high level of adoption of preventive measures. The mean number of practices performed per person was 8.4 of 12, with 87% reported performing at least 7 of the 12 preventive practices. The least practiced were using a mask when cleaning, using gloves when cleaning, keeping trash in closed, rodent-resistant containers, and sealing holes and cracks in the home. All four require buying materials, which might explain why these practices have the lowest percentage of people performing them in this community.

Within the health belief model elements, most were worried that they or their family would get sick with hantavirus (63%, 80%, respectively). This is similar to McConnell's findings, as 73% said they were worried or very worried. With the high number of cases in the community of El Bebedero and most participants knowing at least one person who has been sick with hantavirus, we expected worry to be slightly higher. However, 94% of the sample said they recognized hantavirus as a "dangerous" or "very dangerous" illness. The community understands that hantavirus is a serious disease.

Overall, the community believed the listed preventive practices to be effective and that most were easy to carry out. However, some said they found taking out trash and keeping the home clean as well as keeping waste items such as wood away from the home difficult, mostly due to physical or health restrictions. In future interventions, health officials should consider how they might be able to help residents carry out more physically demanding cleaning practices or move large items such as piles of wood that might serve as refuges for the mouse reservoir.

Only half of respondents said that other residents in their community remind them to carry out the prevention measures. Given that most people found out about hantavirus through other community members and most participants knew someone who had gotten sick with hantavirus, increasing community participation in hantavirus interventions and communications may be an effective strategy. Community participation for infectious disease control has been widely documented, mostly for vector borne diseases such as Chagas, Zika, Dengue. These community-based programs have decreased density of vectors and transmission risk [26–29]. For example, the program *patios limpios* (clean backyards) trained local volunteers to clean up mosquito breeding sites in an organized manner [28]. A similar approach might be taken for hantavirus in Tonosí. Because 86% of respondents said they trust health care professionals the most to receive health information from, a possible approach would be to integrate them into community teams.

In the Spearman correlation matrix, practice score was weakly positively correlated with education and income and negatively correlated with perceived obstacles. However, when controlling for other variables in the multiple linear regression, perceived obstacle score was the only variable found to contribute significantly to the variation in practice score. None of the other elements of the Health Belief Model were found to contribute significantly to the model. For each unit increase in perceived obstacles score (how difficult a participant felt a preventive practice to be), the practice score decreases, which is intuitive and follows the theory proposed by the Health Belief Model. This suggests that in this community, applying interventions to remove barriers to certain preventive practices may increase their uptake. To date, no other Hantavirus studies have used the Health Belief Model to identify predictors of preventive practices. However, the HBM has been used to predict practices for mosquito breeding site control to slow transmission of vector-borne disease such as dengue. For example, using a community-based cross-sectional study in Karachi, Pakistan, Siddiqui et al. found that perceived threat (combination of perceived severity and susceptibility), self-efficacy (belief that one can perform a practice), and knowledge of dengue predicted preventive practices [20]. Like our results, a mixed-methods study by Elsinga et al. found that intention to perform mosquito breeding site control practices were associated with lower perceived barriers in Curacao. However, the barriers identified in the Elsinga study were general such as not knowing how to control breeding sites or feeling as if the government does not help with the control [9]. In a community of indigenous peoples in Malaysia, Chandren et al. also found low perceived barriers to be a predictor of proper prevention practices. Unlike our study, they also found higher knowledge and perceived susceptibility to predict practices [19].

When asked why they thought others in the community did not perform preventive practices, many responded that they either just did not feel like doing them or they did not

"believe" in hantavirus or did not believe the mouse transmitted the virus. Instead, many cited that community members believe chemicals used in crop fumigations of fields that border the homes are the cause of hantavirus respiratory symptoms. This idea has circulated for many years. In mid-2018, a group of protestors in Tonosí blocked the main road due to the increase in hantavirus cases, seeking clarification on whether the mouse or the fumigations were to blame [30]. However, no studies have been conducted to support the link between hantaviruses cases and crop fumigations.

The ordinal logistic regression revealed that each one-year increase in education increases the odds of having a higher hantavirus knowledge score by 1.27 times. Additionally, each increase in person known who had hantavirus would increase odds of higher knowledge by 1.1 times. There are currently no studies published that attempt to identify predictors for knowledge level for hantavirus. Similar studies exploring knowledge predictors for other infectious diseases in Latin America are also scare. However, in a Panama City KAP study for mosquito-borne viruses, Whiteman et al. found that older participants (over 70) had lower knowledge scores than younger participants. They also found that those in higher income brackets had higher knowledge scores. They did not find education to be a predictor of knowledge score for these diseases[13]. In our study community, the results suggest that future campaigns should be aimed at those with lower level of education. Additionally, because the model suggests that knowing more people who have been sick with hantavirus predicts higher knowledge score, future interventions should actively involve those who have experienced the disease as information disseminators. This community is likely more receptive to information given by personal testimonies than television announcements and flyers.

It is important to consider these results in a deeper socioeconomic context. Many homes in El Bebedero border rice and corn fields, the habitat of the hantavirus reservoir. Even with proper prevention measures, the mouse reservoir can still approach the homes due to proximity [31]. Additionally, about 22% of the population in El Bebedero live in poverty [32]. Economic restrictions may make buying proper cleaning supplies difficult, like masks and rubber gloves or materials to seal cracks and holes in walls so that mice cannot enter. Additionally, at least 15% of homes have dirt floors [33], which make it difficult to perform preventive practices, such as mopping instead of sweeping as to not raise dust. While there are still knowledge gaps about route of transmission in the community, providing more information is insufficient to change practices alone. To be successful, future interventions should be grass-roots community organized and consider restricted resources of residents.

Limitations to this study include the use of convenience sample for recruitment. Though we interviewed throughout the day and in the evening, those were not home during specific hours may have represented different socioeconomic groups. For example, if those who were at work during the hours we interviewed represented those who had a higher education level and higher knowledge of hantavirus, this may have lowered the median knowledge score of the sample. Ideally, cluster-random sampling from a pre-created census of the community would have been used. Additionally, because our sample was not a simple random sample, our sample size calculation was inappropriate. However, we did proportionally represent the different neighborhoods within the corregimiento. Due to lack of a true random sample, one should take caution in interpreting the results of the regression models. Without true random sampling, the independence assumption of linear regression may have been violated. We have included these regression models alongside the descriptive analyses to provide insight into important relationships that can be explored more thoroughly in larger, randomized studies. While the study might not be generalizable to other areas in Tonosí, the descriptive statistics given above may be used to inform interventions designed in the population studied (large neighborhoods in El Bebedero). Another limitation that is present in almost all KAP surveys is

response bias. Practices were self-reported and bias toward socially desirable responses and may exist. For preventive practices, we tried to correct for this by awarding more points to spontaneously reported practices.

## Conclusion

Our study has shown that while familiarity and knowledge about hantavirus is relatively high in the region, there is still a knowledge gap with respect to route of transmission. The community seems to understand the danger associated with the virus and many are worried that they or their families will become ill with the disease. Rate of self-reported preventive practices were high for most practices except for wearing gloves and masks while cleaning, keeping trash in rodent-resistant containers, and sealing cracks in the home. Some reported physical restrictions as barriers to cleaning patios and moving accumulations of waste materials, which should be considered in future interventions. Since most respondents had heard of hantavirus through community members, future interventions should be community-based and involve those who have experienced the disease as information disseminators. Future education should also be focused on those with a lower education level.

## Supporting information

**S1 Checklist. STROBE Checklist.**
(DOC)

**S1 Data. Frequency of responses for survey questions.** Practice responses are not included here as they are provided in the main text.
(DOCX)

## Acknowledgments

The authors express their gratitude to all team members in the Department of Investigation of Emerging and Zoonotic Disease at the Gorgas Memorial Institute for Health Studies. A special thank you to the field-interviewers Jacqueline Salazar, Evelyn Rodriguez, Dayana Menchaca, and Enós Juárez, to information systems programmer Claudia Domínguez for configuring the survey tablets, and Publio González for providing maps of the surveyed area. Additionally, this project would not have been possible without the support of Rosa Enith Carrillo de Vargas, Iris Reyes, José Miguel Montenegro, and Pablo Gutiérrez. We are so grateful for the support and guidance of Leisy Villarreal and Lorenzo Aldobán of Tonosí.

## Author Contributions

**Conceptualization:** Carlyn Harris, Blas Armién.

**Formal analysis:** Carlyn Harris.

**Funding acquisition:** Blas Armién.

**Investigation:** Carlyn Harris.

**Methodology:** Carlyn Harris, Blas Armién.

**Project administration:** Carlyn Harris, Blas Armién.

**Resources:** Blas Armién.

**Supervision:** Blas Armién.

**Writing – original draft:** Carlyn Harris.

**Writing – review & editing:** Blas Armién.

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
