## [Decision Letter · Decision Letter 0]

31 Aug 2019

Dear MD Armién:

Thank you very much for submitting your manuscript "Sociocultural determinants of adoption of preventive practices for hantavirus: A knowledge, attitudes, and practices survey in Tonosí, Panama" (#PNTD-D-19-01059) for review by PLOS Neglected Tropical Diseases. Your manuscript was fully evaluated at the editorial level and by independent peer reviewers. The reviewers appreciated the attention to an important problem, but raised some substantial concerns about the manuscript as it currently stands. These issues must be addressed before we would be willing to consider a revised version of your study. We cannot, of course, promise publication at that time.

We therefore ask you to modify the manuscript according to the review recommendations before we can consider your manuscript for acceptance. Your revisions should address the specific points made by each reviewer. 

When you are ready to resubmit, please be prepared to upload the following:

(1) A letter containing a detailed list of your responses to the review comments and a description of the changes you have made in the manuscript.

(2) Two versions of the manuscript: one with either highlights or tracked changes denoting where the text has been changed (uploaded as a "Revised Article with Changes Highlighted" file); the other a clean version (uploaded as the article file).

(3) If available, a striking still image (a new image if one is available or an existing one from within your manuscript). If your manuscript is accepted for publication, this image may be featured on our website. Images should ideally be high resolution, eye-catching, single panel images; where one is available, please use 'add file' at the time of resubmission and select 'striking image' as the file type. 

Please provide a short caption, including credits, uploaded as a separate "Other" file. If your image is from someone other than yourself, please ensure that the artist has read and agreed to the terms and conditions of the Creative Commons Attribution License at http://journals.plos.org/plosntds/s/content-license (NOTE: we cannot publish copyrighted images). 

(4) If applicable, we encourage you to add a list of accession numbers/ID numbers for genes and proteins mentioned in the text (these should be listed as a paragraph at the end of the manuscript). You can supply accession numbers for any database, so long as the database is publicly accessible and stable. Examples include LocusLink and SwissProt.

(5) To enhance the reproducibility of your results, we recommend that you deposit your laboratory protocols in protocols.io, where a protocol can be assigned its own identifier (DOI) such that it can be cited independently in the future. For instructions see http://journals.plos.org/plosntds/s/submission-guidelines#loc-methods

While revising your submission, please upload your figure files to the Preflight Analysis and Conversion Engine (PACE) digital diagnostic tool, https://pacev2.apexcovantage.com/ PACE helps ensure that figures meet PLOS requirements. To use PACE, you must first register as a user. Then, login and navigate to the UPLOAD tab, where you will find detailed instructions on how to use the tool. If you encounter any issues or have any questions when using PACE, please email us at figures@plos.org.

We hope to receive your revised manuscript by Oct 30 2019 11:59PM. If you anticipate any delay in its return, we ask that you let us know the expected resubmission date by replying to this email.

To submit a revision, go to https://www.editorialmanager.com/pntd/ and log in as an Author. You will see a menu item call Submission Needing Revision. You will find your submission record there. 

Sincerely,

Colleen B Jonsson, PhD

Guest Editor

Scott Weaver

Deputy Editor

Reviewer's Responses to Questions

Key Review Criteria Required for Acceptance?

Methods

-Are the objectives of the study clearly articulated with a clear testable hypothesis stated?

-Is the study design appropriate to address the stated objectives?

-Is the population clearly described and appropriate for the hypothesis being tested?

-Is the sample size sufficient to ensure adequate power to address the hypothesis being tested?

-Were correct statistical analysis used to support conclusions?

-Are there concerns about ethical or regulatory requirements being met?

Reviewer #1: The methods are all relevant and the analysis is well done.

Reviewer #2: Yes to first 5.

No ethical concerns. Study approved by appropriate IRB and informed consent obtained.

Authors state that original data can't be released due to confidentiality concerns.

Reviewer #3: The authors need to explicitly state that the seven neighborhoods in the study were purposefully selected based on their large population size; meaning that smaller neighborhoods were excluded from the study. Therefore the results are biased toward the larger communities and not representative of El Bebedero. It is important to state this upfront because it has implications for how the results are interpreted. Because households and individuals were also conveniently selected, then it’s evident that random selection did not occur at any level. I’m curious to know how the authors went from neighborhood selection to household selection? Did data collection teams go to every household in each neighborhood to give them the chance to be included in the survey? Or was an intermediary step used (i.e. selection of clusters within neighborhoods) ?

I ask these questions, because when doing a KAP survey, or any survey for that matter, it is very important to be able to know who the findings are ‘valid’ for (i.e. external validity). In this case, your results may only be potentially applicable to just those respondents in the 7 neighborhoods whose households were conveniently selected and had someone at home at the time of the visit who was willing to provide consent. As you can see there are many caveats here in terms of who the findings may be applied to.

The sample size calculation assumptions described by the authors are based on having a sampling frame of all individuals in Bebedero and using simple random sampling to select individuals. The sample size obtained from this is usually called the effective sample size (ESS) or the minimum required sample when using simple random sampling. Apparently simple random sampling was not their sampling approach used so therefore the sample size calculations are incorrect because they ignore other important parameters. At a minimum, they needed to have accounted for the intraclass correlation (ICC) which would have undoubtedly increased the ESS because the design effect would likely have been at least 2 or more; meaning that ESS of 95 needed to be multiplied by at least 2, which would’ve meant about 200 respondents plus inflation for non-response (200 divided by response rate; e.g. 200/.80 = 250). In sum, it would have been more appropriate to just use multi-stage cluster sampling, and then account for the cluster design using inverse probability weights for each level of selection. State this somewhere in the limitation section. 

The section on consent is a bit confusing. Consent was sought after residents agreed to participate? (lines 164-165: If a resident agreed to participate in the survey, they read the informed consent and were allowed to ask questions). I’m assuming this was a translation/language issue. However, need to review and be sure to have it reflect what was done. Normal procedure is to explain the purpose of the study, its risks, benefits, etc (consent statement) then seek informed consent from the participant.

Because of the limitations posed by the sampling, the authors need to recognize that they are limited in the type of analysis that can be conducted without violating underlying assumptions of the statistical tests used. Therefore, while I generally appreciate the use of regression models, I think that the sample was not powered appropriately for these hypothesis testings. Therefore, my recommendation is for the authors to focus on the descriptive results (those currently included in the supplemental material) with the understanding that the results are not generalizable but formative. 

The regression models are inappropriate for use in this study, and any conclusions drawn from them would be misleading because the standard errors are under-estimated by not accounting for the intraclass correlation (which violates the assumption of independence in regression modeling). In essence the statistical software is ‘thinking’ that the sample was obtained from simple random sampling from a population-based sampling frame (which obviously is not the case) when determining significance in testing the hypotheses. Therefore, the null may have been rejected when it shouldn’t have because of the of the incorrect standard errors. So things that are significant are either (1) not significant or (2) the sample size is not large enough to know if they are in fact significant. Therefore, the most the authors can do is to describe results using proportions and aggregated mean KAP scores, etc. Because of the limited evidence available on the topic they have presented on, the descriptive results still have good public health value. In a way, the authors should view this as an early exploratory assessment (formative research) to inform future KAP surveys on hantavirus that will use more rigorous sampling approaches and complex analyses in the same population or similar populations elsewhere. 

The big take-away here is that when conducting a KAP survey, the sampling approach determines how the analysis can be conducted. Fancy analytical approaches cannot override sub-par sampling techniques and lack of statistical power. Convenience sampling should only be done when a random sample is impossible to obtain due to specific characteristics of the population being targeted (e.g. hidden population without good sampling frame), or when wanting to just get some quick/rapid ‘impressions/insights’ to inform future work (the current paper falls in this category in my opinion and should be framed as such). They authors should not shy away from what they’ve done but rather learn from it and embrace its limitations as a stepping stone in the right direction for their future work and that of others in their field.

Results

-Does the analysis presented match the analysis plan?

-Are the results clearly and completely presented?

-Are the figures (Tables, Images) of sufficient quality for clarity?

Reviewer #1: The analysis match the analysis plan and the results are clearly presented, and in general the tables/figures are of sufficient quality

However, I have some questions.

Was there any analysis on results depending on gender? Or on occupation?

Please explain all abbreviations in the tables under each table, the tables should be able to stand for themselves.

Furthermore, to make the tables clearer, please modify as suggested below.

Reviewer #2: Yes to all. Information easy to follow. If space does not allow printed all 7 tables, perhaps some could be included in an appendix.

Reviewer #3: As mentioned in my comments regarding the methods, the analysis need to focus on describing the results using what they already have included in the suppl material. However, it won’t be realistic to include all the suppl tables. Instead the authors must try to find ways to combine the tables and/or summarize the results a bit more and keep just the key tables and leave the rest as suppl.

Conclusions

-Are the conclusions supported by the data presented?

-Are the limitations of analysis clearly described?

-Do the authors discuss how these data can be helpful to advance our understanding of the topic under study?

-Is public health relevance addressed?

Reviewer #1: The conclusions supported by the data presented, limitations are clearly described, the authors discuss how these data can be helpful to advance our understanding of the topic under study and public health relevance is addressed.

Reviewer #2: Yes to all. Authors specifically detail limitations of the survey methods and possible response bias.

Reviewer #3: The conclusion would need to be revised based on revisions to the presentation of the results. Generally, the conclusions for this assessment should be formative – i.e. trying to inform future areas of work and/or focus regarding assessments of hantavirus infection prevention at the community level in the absence of biomedical treatment and vaccines.

Editorial and Data Presentation Modifications?

Reviewer #1: Minor modifications as outlined below are needed. I suggest a Minor Revision.

Introduction.

Page 4, line 70. The bunyavirus taxonomy has recently been updated. Hantaviruses now have their own family, Hantaviridae. (ICTV. https://talk.ictvonline.org/taxonomy/)

The order Bunyavirales now contain 12 families, one of them Hantaviridae, with the subfamily Mammantavirinae, genus Orthohantavirus. But it is enough to change “Bunyaviridae” to “Hantaviridae” in line 70.

When first mentioned please add the abbreviation directly after the full description. For instance:

Line 74 and 75: Hantavirus Fever with Renal Syndrome (HFRS) and Hantavirus Pulmonary

Syndrome (HPS). 

Line 81: Here a new name appear: “..cardiopulmonary” , please decide on one description, and use the abbreviations when you have introduced them.

Line 86. The mortality rate is here stated as being 16.5% - is there a reference for that?

And is it really the mortality rate? Is it not rather the case-fatality rate, which usually is higher than the mortality rate. Also, the authors report a case-fatality rate of 4/103=4% in 2018 in the Azuero region (line 89). How does this 4% case-fatality rate relate to the 16.5% mortality rate on line 86? In Armien et al., Am J Trop Med Hyg. 2013, 89:489-94, they write about “the high frequency of HV fever and mild HPS in Choclo virus infection…”, which may indicate that 16.5% mortality may be an overestimation.

Line 91: Add “way” to the sentence: “Preventive measures to avoid human contact with rodents and their excrements are the most effective WAY to prevent…”

Line 121-122: The sentence is not clear “It was predicted that at least one of the Health Belief Model elements (see below) and level of knowledge of hantavirus would predictors of preventive practice“.

Could it instead be: “It was predicted that at least one of the Health Belief Model elements (see below) and the level of knowledge of hantavirus would be predictors of preventive practice.”

Results: 

In general. Was there any analysis on results depending on gender? Or on occupation?

In general: Please explain all abbreviations in the tables under each table, the tables should be able to stand for themselves.

Lines 253-254 and line 258.

First the authors show that 71% (60.5% + 10.5%) correctly recognized the mouse reservoir, and then 86% (line 258)? Please explain.

Lines 285-287. Figure 1. Is this figure really needed? The same info is in table 2. If needed, then the figure legend needs improvement, explain all abbreviations in the figure and in the legend title it should be obvious that this is after prompting.

Lines 366-367. Table 5. Please write the numbers with a 0 before the point. That is, write 0.156 instead of .156 and so on in the whole table.

Line 372. Here the authors use the greek beta (β), in table 6 they write Beta, please be consistent.

Lines 376-377. Table 6. Please write the numbers with a 0 before the point. That is, write 0.031 instead of .31 and so on in the whole table.

Line 387, change “and” to “an”

Line 389-391. This sentence needs grammatical input, and perhaps change “ill with Hanta” to “hantavirus disease” or “HFRS” or something similar.

Lines 394-395. Table 7. Please write the numbers with a 0 before the point. That is, write 0.006 instead of .006 and so on in the whole table.

Discussion

Line 404. Regarding the discussion on “crop fumigations near homes are linked to hantavirus”. Why is it like this? Do the communities get similar disease symptoms as hantavirus disease when there is crop fumigations? Or does the crop fumigations clear out all rodents, and then some time after, new virus-carrying rodents can appear in the clean environment? Are there some refs on this matter? Is it rodent fumigations or insect fumigations?

Lines 489-490. What was the result of the “clarification on whether the mouse or the fumigation was to blame”?

Could fumigations of crop drive the mouse away from the crops to seek protection indoors, and thereby increase the risk of close-contact with the community?

Reviewer #2: Minor spelling and usage concerns:

Lines 120-24: excessive use of "predict" ; insert "be" between "hantavirus would" and "predictors" in lines 121-22.

158: misspelling of "corregimiento"

160: drop "of" after "comprised"

365 and 462: should be "Spearman"

371-72: should be" found to make an independent statistically significant contribution"

416: "Effected" should be "Affected"

Reviewer #3: Major revisions needed regarding presentation of the results.

Summary and General Comments

Reviewer #1: (No Response)

Reviewer #2: Recommend acceptance with minor revisions. None of the findings were earth-shaking or entirely unexpected but should be useful in helping to reduce the impact of hantavirus in Panama and socioculturally similar areas of the region and the world.

Reviewer #3: The authors have addressed an important research area focusing on the sociocultural determinants of hantavirus prevention practices and knowledge among community members in a highly affected geographic area in Panama prone to the hantavirus infections. Given that biomedical treatments and vaccines are not available to tackle this pathogen, community-based prevention practices are critical for disease control. Therefore, this KAP study address an issue with public health importance for a specific population and potential implications to inform other studies elsewhere. 

In the background, instead of saying “we predicted” it’s better to say “hypothesized” because you predict after collecting data and hypothesized before that.

It’s also better to say that the Health Belief Model informed the design of the KAP instead of saying that the survey was based on HBM. Basing the survey on HBM will require psychometric validation of the HBM constructs used in the study, which to my understanding was not done. Also, note that cues-to-action is not an attitude and should not be described as such. Self-efficacy is an important construct in HMB but missing from this study. How confident people are in their ability to execute the behavior is strong predictor of the likelihood of behavior uptake. The authors therefore need to explain the omission of self-efficacy in their use of HBM. In addition, there are many other individual health behavior models, why HBM? Need to justify the use of HBM in the context of the alternatives. Using theory to ground a KAP survey requires a lot of steps that are currently missing. Therefore, the authors may want to de-emphasize HBM as a grounding theory and more so as a theoretical guidance coupled with the prior literature on KAPs on hantavirus. In the background, a more synthesized summary of HBM should be sufficient, and then in the methods the authors can elaborate on the specific adaptations for their study.

The regression models are inappropriate for use in this study, and any conclusions drawn from them would be misleading because the standard errors are under-estimated by not accounting for the intraclass correlation (which violates the assumption of independence in regression modeling). However, there are some really interesting results in the suppl material that can be synthesized nicely in the results section to tell an important story about the potential barriers and enablers of hantavirus prevention practices in this highly affected population.

PLOS authors have the option to publish the peer review history of their article (what does this mean?). If published, this will include your full peer review and any attached files.

Do you want your identity to be public for this peer review? For information about this choice, including consent withdrawal, please see our Privacy Policy.

Reviewer #1: No

Reviewer #2: No

Reviewer #3: Yes: Mohamed F Jalloh

---

## [Decision Letter · Decision Letter 1]

3 Feb 2020

Dear MD Armién,

We are pleased to inform you that your manuscript 'Sociocultural determinants of adoption of preventive practices for hantavirus: A knowledge, attitudes, and practices survey in Tonosí, Panama' has been provisionally accepted for publication in PLOS Neglected Tropical Diseases.

Before your manuscript can be formally accepted you will need to complete some formatting changes, which you will receive in a follow up email. A member of our team will be in touch within two working days with a set of requests.

Best regards,

Colleen B Jonsson, PhD

Guest Editor

Scott Weaver

Deputy Editor

Reviewer's Responses to Questions

**Key Review Criteria Required for Acceptance?**

**Methods**

-Are the objectives of the study clearly articulated with a clear testable hypothesis stated?

-Is the study design appropriate to address the stated objectives?

-Is the population clearly described and appropriate for the hypothesis being tested?

-Is the sample size sufficient to ensure adequate power to address the hypothesis being tested?

-Were correct statistical analysis used to support conclusions?

-Are there concerns about ethical or regulatory requirements being met?

Reviewer #2: Limitations of the analysis, primarily due to the lack of a true random sample have now been specifically acknowledged in the paper.

Reviewer #3: The authors have adequately address previous comments on the methods.

**Results**

-Does the analysis presented match the analysis plan?

-Are the results clearly and completely presented?

-Are the figures (Tables, Images) of sufficient quality for clarity?

Reviewer #2: Yes to all

Reviewer #3: The authors have adequately address previous comments on the results.

**Conclusions**

-Are the conclusions supported by the data presented?

-Are the limitations of analysis clearly described?

-Do the authors discuss how these data can be helpful to advance our understanding of the topic under study?

-Is public health relevance addressed?

Reviewer #2: Yes to all.

Reviewer #3: The authors have adequately address previous comments on the conclusion.

**Editorial and Data Presentation Modifications?**

Reviewer #2: Minor grammatical changes are needed, but nothing major has been identified.

Line 436: "sparse" is a better term than "scarce" here.

Line 467: "most were worried that themselves or their family" should be "they or their family"

Line 545: "information in insufficient" should be "information is insufficient"

Reviewer #3: The table labels may need to be revised for simplicity and clarity, especially Table 7.

**Summary and General Comments**

Reviewer #2: Given that the author has fully acknowledged the limitations of the study due to the lack of a true random sample, I believe that the findings, though preliminary in nature, are worthy of sharing with the scientific community

Reviewer #3: The authors have done a stellar job of addressing the feedback from the last round of review. I don't have any additional comments.

PLOS authors have the option to publish the peer review history of their article (what does this mean?). If published, this will include your full peer review and any attached files.

Reviewer #2: No

Reviewer #3: No

---

## [Editor Report · Acceptance letter]

25 Feb 2020

Dear MD Armién,

We are delighted to inform you that your manuscript, "Sociocultural determinants of adoption of preventive practices for hantavirus: A knowledge, attitudes, and practices survey in Tonosí, Panama," has been formally accepted for publication in PLOS Neglected Tropical Diseases.

Best regards,

Serap Aksoy

Editor-in-Chief

Shaden Kamhawi

Editor-in-Chief
